# Should Magnetic Resonance Angiography Be Used for Screening of Intracranial Aneurysm in Adults with Sickle Cell Disease?

**DOI:** 10.3390/jcm11247463

**Published:** 2022-12-16

**Authors:** Igor Gomes Padilha, François Guilbert, Laurent Létourneau-Guillon, Stéphanie Forté, Kristoff Nelson, Manon Bélair, Jean Raymond, Denis Soulières

**Affiliations:** 1Radiology Department, Centre Hospitalier de l’Université de Montréal (CHUM), Université de Montréal, Montreal, QC H3T 1J4, Canada; 2Haematology/Oncology Department, Centre Hospitalier de l’Université de Montréal (CHUM), Université de Montréal, Montreal, QC H3T 1J4, Canada

**Keywords:** intracranial aneurysm, sickle cell disease, MR imaging, MR angiography

## Abstract

Magnetic resonance imaging (MRI) is used in patients with sickle cell disease (SCD) to detect silent cerebral infarcts. MR angiography (MRA) can identify arterial stenoses and intracranial aneurysms (ICANs) associated with SCD. In this study, we aimed to estimate the prevalence of ICANs in asymptomatic adult patients with SCD referred from the SCD clinic for routine screening by MRI/MRA using a 3T-MRI scanner. Findings were independently reviewed by two neuroradiologists. Between 2016 and 2020, 245 asymptomatic adults with SCD were stratified according to genotype (SS/S-β0thalassemia and SC/Sβ+). ICANs were found in 27 patients (11%; 0.95 CI: 8–16%). ICANs were more frequent in SS/S-β0thalassemia patients (20/118 or 17%; 0.95 CI: 11–25%) than in SC/βb+ patients (7/127 or 6%; 0.95 CI: 2–11%; *p* = 0.007). Individuals with SCD (particularly SS/S-β0thalassemia) have a higher prevalence of ICANs than the general population. We believe that MRA should be considered in the current American Society of Hematology guidelines, which already contain a recommendation for MRI at least once in adult SCD patients. However, the clinical significance of preventive treatment of unruptured aneurysms remains controversial.

## 1. Introduction

Sickle cell disease (SCD) is associated with vaso-occlusive phenomena which can result in a wide spectrum of clinical manifestations [1,2,3,4]. Neurovascular complications such as ischemic or hemorrhagic strokes, symptomatic or silent, may occur in up to one in four patients over their lifetimes [1,2,3]. Transcranial Doppler (TCD) has been adopted to identify patients at risk of an ischemic event [5,6]. However, TCD does not select patients at risk of hemorrhagic events, which are known to occur later in the evolution of the disease (usually third and fourth decade of life) [7].

MRI screening has been recommended by the American Society of Hematology (ASH) to detect silent cerebral infarctions (SCI) associated with neurocognitive impairment in a significant proportion of patients [4,8,9] MR angiography (MRA) performed at the time of screening for SCI may reveal arterial stenoses or intracranial aneurysms (ICANs) [10,11]. The recently reported POST-STOP cohort evaluated retrospectively an incidence of hemorrhagic stroke of 63 per 100,000 person years overall and 134 for adults [12]. Vascular abnormalities were noted in 18/35 patients with hemorrhagic stroke. This study only reports results for SS/S-β0thalassemia patients. Adult outpatients with SCD, including SS/S-β0thalassemia/Sβ+ and SC genotypes, at our institution are routinely surveilled for vascular diseases with 3-dimensional time-of-flight MR angiography at 3T.

In this study, we aimed to estimate the prevalence of ICANs in all patients with SCD screened by MRA over a period of 5 years and the effect of genotype.

## 2. Materials and Methods

This retrospective study was approved by the research ethics committee of the Centre Hospitalier de l’Université de Montréal and the requirement for informed consent was waived. Adult outpatients followed by the SCD clinic of the Department of Hematology were included if one or more 3T MRI/ MRA studies were performed as part of a screening program. Patients were categorized according to genotype: SS/S-β0thalassemia and SC/S-β+ Graham-Serjeant type 1. Inclusion criteria for this study were: age ≥ 18, genotypically confirmed SCD, and no prior history of neurovascular complications and at least one yearly follow-up.

All MRIs were performed on a 3T scanner (Philips Ingenia, Philips Healthcare, Best, The Netherlands), and included three-dimensional fluid-attenuated inversion recovery (FLAIR) and time-of-flight intracranial angio-MRA without gadolinium injection. All studies were reviewed by a fellow in diagnostic neuroradiology and a certified neuroradiologist with more than 20 years of experience. Two independent readings were carried out and the discrepancies were resolved by consensus. The management and follow-up of patients with asymptomatic aneurysms were recorded.

The prevalence of ICANs along with 95% confidence intervals (calculated according to Exact Clopper-Pearson method) for each genotype was compared using Fisher’s exact test (StatsDirect, Version 3.3.4, Cambridge, UK).

## 3. Results and Discussion

Between 1 January 2016, and 31 December 2020, 271 patients with SCD were referred for brain MRI/MRA. This population was followed regularly and 84% were receiving a treatment (hydroxyurea 68%, transfusion program 16%). Twenty-two patients were excluded because of neurological symptoms and four other patients because of non-diagnostic imaging, leaving 245 neurologically asymptomatic adults with SCD studied over 5 years with a total of 369 brain MRI/MRAs (Figure 1).

The median age (interquartile range) was 32 (25–42) years, and 125 (51%) were women. Unruptured saccular aneurysms were found in 27 patients (11%; 0.95 CI: 8–16%). Nine patients (33%) had more than one aneurysm, for a total of 44 aneurysms: 40 were in the anterior circulation and four in the posterior circulation. All aneurysms were ≤10 mm (median 2.5 mm) (Table 1).

One patient with an asymptomatic 10 mm aneurysm was treated by coiling. Everyone else was observed, including 17 patients with asymptomatic aneurysms who underwent at least one follow-up MRA with a median time of 13.5 months between studies. One patient with a 3 mm internal carotid aneurysm also had a middle cerebral artery stenosis.

Aneurysms were more frequent (*p* = 0.007; Fisher’s exact test) in SS/S-β0thalassemia patients (20/118 or 17% 0.95 CI: 11–25%) than in SC/Sβ+ patients (7/127 or 6%; 0.95 CI: 2–11%). More specifically, in SC patients the prevalence was 3/110 or 3% 0.95 CI: 1–8%). Aneurysms were also more frequent (*p* = 0.03; Fisher’s exact test) in patients over 30 (15%; 0.95 CI: 10–22%) as compared to younger patients (6%; 0.95 CI: 3–12%). Median age of individuals with ICANs was 38 years and 43 years in SS/S-β0thalassemia and SC/Sβ+ respectively. Seventeen out of 27 ICAN+ patients underwent ≥ 2MRI/A. All ICANs remained stable in terms of size and morphology during consecutive scans.

To the best of our knowledge this study is the largest cohort of MRI-screened asymptomatic SCD patients and confirms that adult patients with SCD, and particularly SS/S-β0thalassemia, have a higher prevalence of ICANs than the general population [11,13]. The prevalence of asymptomatic aneurysms in the general adult population is approximately 2–3%. One systematic review of 1450 unruptured ICANs identified by different imaging modalities (including MRA) in 94,912 individuals included in 68 studies found a prevalence of 3% (0.95 CI: 2–3.9%) [13]. Other publications, such as a study in 2000 healthy adults that did not include MRA evaluation, have found the prevalence of incidental aneurysms to be closer to 1.8% [11]. The data on ICAN in the general population are based on incidental findings and reports a median age of 63.3 yo (range 47.7–96.7). Our data show a median age of 38 and 43 years depending on genotype. Although the study design is different from the report of prevalence in the general population, data suggest SCD patients appear to be younger at presentation with a potential for more years at risk and morbidity from ICAN complications.

Our results are in line with two previous publications on SCD reporting the prevalence of ICANs in 9% (5/55 patients) [10] and 6% [14] of adult patients. The last study also identified a higher frequency of ICANs in patients with the HbSS/S-β0thalassemia genotype (8%), which was significantly higher than that for the other sickle cell genotypes (1%; *p* < 0.05) [14]. One significant difference to prior studies is the use of a 3T high-field scanner enabling a higher resolution for vascular imaging. This particularity potentiates the detection of small aneurysms, thereby increasing the sensitivity, a possible explanation for the increased prevalence compared to previous reports [15,16], although the clinical significance of identifying these smaller aneurysms and subtle vascular anomalies remains uncertain. The authors of the POST-STOP retrospective cohort analysis proposed that the results warrant screening for actionable cerebrovascular abnormalities, based on the high incidence of hemorrhagic stroke and underlying vascular anomalies in a population considered at risk based on the inclusion criteria of STOP I and STOP II [12].

According to the American Heart/American Stroke Association, screening for ICAN is only recommended for patients at increased risk of aneurysm development, such as patients with a positive family history or those with predisposing disorders such as autosomal dominant polycystic kidney disease, coarctation of the aorta, or rare collagen diseases. Interestingly, all the above-mentioned groups show a frequency of ICANs comparable to our cohort of SCD patients [17]. None of these screening recommendations are based on a proof of clinical benefits of treating unruptured aneurysms [18].

MRA in the setting of unruptured ICAN screening is well recognized. A systematic review and meta-analysis reported a sensitivity of MRA in the diagnosis of ruptured and unruptured ICANs (960 patients assessed, 772 ICANs) of 95% (0.95 CI, 89–98%) [19] in comparison with another study that described a sensitivity of 45% of ICANs detected on conventional MRI sequences after evaluating 45 patients (57 ICANs) initially diagnosed with unruptured ICANs on CTA that had brain MRI within 6 months before CTA [20]. Additionally, the increased potential of MRA to identify unruptured aneurysms remains unclear to prevent the clinical consequences or warrant preventive therapy, especially in this specific population. This will be the object of future validation cohorts and prospective longitudinal follow-up studies.

This is a monocentric study with a total number of patients that remains small, though much larger than previously published cohorts. Risk factors for ICANs, such as family history of an aneurysm, smoking or hypertension, were not systematically recorded as potential confounding factors as in causal research. Other factors associated with SCD specifically will be assessed in a future study addressing not only prevalence, but also incidence of ICANs, namely disease-modifying treatments such as hydroxyurea and transfusion program, ∝-globin deletions, SCD non-neurologic events and rate of SCD events.

## 4. Conclusions

In summary, patients with SCD, and particularly SS/S-β0thalassemia patients, appear to have a higher prevalence of intracranial aneurysms than the general population. These results suggest that MRA could be added to the American Society of Hematology screening recommendations, which already include a routine MRI at least once in adults with SCD. However, the clinical significance and preventive treatment of unruptured aneurysms remain controversial and requires further inquiry. Future studies should aim to validate the results from our testing cohort. Additionally, longitudinal outcomes need to be assessed to better determine the long-term impact of these lesions. The natural history and real impact of specific treatments is not well known in literature. Future prospective and populational studies will help to define and evaluate specific surveillance and therapeutic strategies in this population.

## Figures and Tables

**Figure 1 jcm-11-07463-f001:**
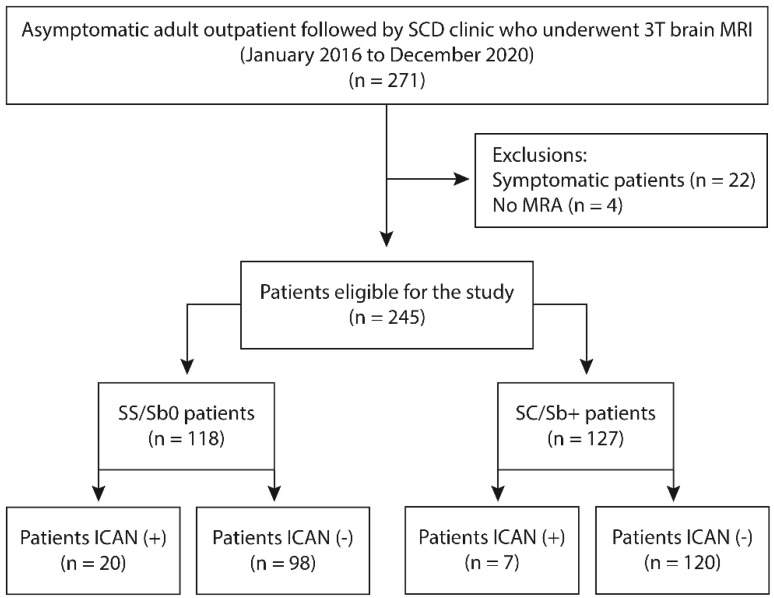
Flowchart of patients included in the study. SCD: Sickle cell disease; MRI: Magnetic resonance imaging; ICAN: intracranial aneurysm; SS/Sb0 = SS/S-β0thalassemia genotype; SC/Sb+ = SC/S-β+ genotypes.

**Table 1 jcm-11-07463-t001:** Size and location of 44 ICAN in 27 patients with SCD.

Size	ICA	MCA	ACA	ACoA	Posterior Circulation	Total
<3 mm	18	1	1	0	1	21
3–5.9 mm	15	1	1	0	3	20
6–10 mm	1	1	0	1	0	3
>10 mm	0	0	0	0	0	0
Total	34	3	2	1	4	44

ICAN: intracranial aneurysm; SCD: Sickle cell disease; ICA: Internal carotid artery; MCA: Middle cerebral artery; ACA: Anterior cerebral artery; ACoA: Anterior communicating artery.

## Data Availability

After the selection of clinical data the radiological reports were consulted and the relevant information collected in a database which kept by the principal investigator on a digital document encrypted and protected by a password. A single copy of this file is kept on the intranet network in a file saved by the CHUM Information and Telecommunications Technology Department.

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
