# Peer review of "Should Magnetic Resonance Angiography Be Used for Screening of Intracranial Aneurysm in Adults with Sickle Cell Disease?"

_jcm, 2022, doi:10.3390/jcm11247463_

Round 1

Reviewer 1 Report

It is an effective, clear study.

The prevalence of ICAN is well determined in this population and this is the largest cohort of MRI screened asymptomatic SCD population. The result are interesting.

As we know that  SCD patients and particularly SS/S-β0thalassemia, have a higher prevalence of ICANs than the general population, what surveillance and therapeutics interventions can be discuss in this population?

This is a monocentric study and it will be of interest to know a little bit of the history of the disease. They have no prior history of neurovascular complications but do they have others complications? … numbers of crisis, nephropathy, cardiopathy... and do they have treatments? . Because it is a monocentric study maybe these informations can be easily found. 

Reviewer 2 Report

The manuscript reports on the single center experience on the use of cranial MRI+MRA in asymptomatic adult patients with SCD with the objective of identifying intracranial aneurysms (ICAN) that require intervention. The main result is the high prevalence of asymptomatic ICANS (11%) that is dependent on the genotype (more frequent in SCD-S/S and -S/ß0thal than in SCD-S/C and -S/ß+thal). The authors compare this prevalence to published data on the general adult population that report a prevalence of ICANs of 2-3 %. 

While the comparison of these numbers provides a clear indication that ICANs are more frequent in patients with SCD than in healthy adults, it is currently not clear if any consequences for the management of patients should be drawn.

The paper is well written and the results are clearly presented. However, I suggest to provide further data that would facilitate the interpretation of the results:

1. Patients with SCD-S/C are predicted to have very little cerebrovascular complications. In this study, they are combined with SCD-S/ß+thal. I suggest to report the prevalence of ICAN separately for SCD-S/C, even if this will be a relatively small group.

2. Patients with SCD-S/ß+ genotype can have little, moderate or high expression HbA (e.g. as classified by Graham Serjeant: type 1 with <7% HbA, type 2 with 7-15% HbA, type 3 with >15% HbA). I expect that patients with type 1 or 2 according to Serjeant will behave like SCD-S/ß0thal or SCD-S/S with respect to (cerebrovascular) complications. Ideally, the prevalence of ICANs would be considered separately in these different genotypes. As this will likely not be possible, I suggest that the precise genotypes of SCD-S/ß0thal are listed. Are these predominantly type 1/2 (behaving like SCD-S/S) or type 3 ?

3. Besides the ß-globin genotype, other genetic traits such as alpha-thalassemia status can modify the risk of cerebrovascular complications. If available, the frequency of these genetic modifiers should be reported.

4. The risk of complications of SCD depends on treatment. Which proportion of patients received hydroxyurea or chronic transfusions?

5. Patients with a history of stroke were excluded. Did any of the patients included in the study have TCD abnormalities in the past? 

6. How many patients with SCD are followed by the authors in total? Was there any selection of patients undergoing MRI/MRA in comparison to the total group of patients with SCD in follow-up? PLease compare the patients undergoing MRI/MRA to all patients in follow-up for the parameters listed in 1-5.

7. In total, 369 cMRA/MRA were done in 245 asymptomatic patients. Was there an algorithm on when a patient should undergo a first, second and third MRI/MRA? Which patients were examined more than once?

8. Are any follow up data available on patients who had an ICAN? Were ICANs at second and third MRI changing in size or quality?

9. The authors point out that the ASH guideines recommend at least one single cMRI in adults with SCD but imply that MRA could increase the sensitivity for detecting ICANs. All patients underwent both MRI and MRA. Can the authors comment on the sensitivity of MRA compared to MRI? How many of the relevant ICAN are already detected by MRI? Are the additional ICAN detected by MRA only relevant or are these the small ones that do not even need follow up? 

10. One patient had a coiling of an ICAN as a consequence of the MRI/MRA. Were any other, non-invasive consequences drawn from the examination? Were patients treated with HU or transfusion as a consequence of the detection of ICAN? How many were examined by DSA in addition to MRA?

11. The prevalence of ICANs appears to be higher in adults with SCD compared to healthy adults. Can this comparison be adjusted for the age distribution in these two groups? I suppose patients with SCD were younger than the reference population, so the difference in the prevalence of ICAN may be much more pronounced after correction for age.

Round 2

Reviewer 2 Report

The authors have sufficiently responded to all suggestions. The manuscript now gives an impression on the frequency of ICANs in adults with SCD and can serve as a basis for future studies that detect risk factors and define management algorithms.

I believe it is important to point out that all patients with SCD-S/ß+thal were "Serjeant type 1" and express only minimal HbA. Consequently, the rate of ICANs was (as far as can be told from the small group) as high in SCD-S/ß+thal as in SCD-S/S.